# Seven Epidemic Waves of COVID-19 in a Hospital in Madrid: Analysis of Severity and Associated Factors

**DOI:** 10.3390/v15091839

**Published:** 2023-08-30

**Authors:** Juan Víctor San Martín-López, Nieves Mesa, David Bernal-Bello, Alejandro Morales-Ortega, Marta Rivilla, Marta Guerrero, Ruth Calderón, Ana I. Farfán, Luis Rivas, Guillermo Soria, Aída Izquierdo, Elena Madroñal, Miguel Duarte, Sara Piedrabuena, María Toledano-Macías, Jorge Marrero, Cristina de Ancos, Begoña Frutos, Rafael Cristóbal, Laura Velázquez, Belén Mora, Paula Cuenca, José Á. Satué, Ibone Ayala-Larrañaga, Lorena Carpintero, Celia Lara, Álvaro R. Llerena, Virginia García, Vanessa García de Viedma, Santiago Prieto, Natalia González-Pereira, Cristina Bravo, Carolina Mariño, Luis Antonio Lechuga, Jorge Tarancón, Sonia Gonzalo, Santiago Moreno, José M. Ruiz-Giardin

**Affiliations:** 1Servicio de Medicina Interna, Hospital Universitario de Fuenlabrada, 28942 Madrid, Spain; nieves.mesa@salud.madrid.org (N.M.); david.bernal@salud.madrid.org (D.B.-B.); alejandro.morales@salud.madrid.org (A.M.-O.); marta.rivilla@salud.madrid.org (M.R.); marta.guerrero@salud.madrid.org (M.G.); ruth.calderon@salud.madrid.org (R.C.); anai.farfan@salud.madrid.org (A.I.F.); luis.rivas@salud.madrid.org (L.R.); guillermo.soria@salud.madrid.org (G.S.); aida.izquierdo@salud.madrid.org (A.I.); elena.madronal@salud.madrid.org (E.M.); miguelangel.duarte@salud.madrid.org (M.D.); saraisabel.piedrabuena@salud.madrid.org (S.P.); mtoledano@salud.madrid.org (M.T.-M.); jorge.marrero@salud.madrid.org (J.M.); cristina.ancos@salud.madrid.org (C.d.A.); begona.frutos@salud.madrid.org (B.F.); rafael.cristobal@salud.madrid.org (R.C.); laura.velazquez@salud.madrid.org (L.V.); belen.mora@salud.madrid.org (B.M.); paula.cuenca@salud.madrid.org (P.C.); joseangel.satue@salud.madrid.org (J.Á.S.); ibone.ayala@salud.madrid.org (I.A.-L.); lorena.carpintero@salud.madrid.org (L.C.); celia.lara@salud.madrid.org (C.L.); alvaroricardo.llerena@salud.madrid.org (Á.R.L.); vgbermudez@salud.madrid.org (V.G.); vggarciadeviedma@salud.madrid.org (V.G.d.V.); sgonzalo@salud.madrid.org (S.G.); josemanuel.ruiz@salud.madrid.org (J.M.R.-G.); 2CIBERINFEC, Instituto de Salud Carlos III, Madrid, 28029 Madrid, Spain; smguillen@salud.madrid.org; 3Departamento de Medicina y Especialidades Médicas, Universidad de Alcalá, 28871 Madrid, Spain; 4Servicio de Laboratorio Clínico, Hospital Universitario de Fuenlabrada, 28942 Madrid, Spain; sprietom@salud.madrid.org (S.P.); ngpereira@salud.madrid.org (N.G.-P.); 5Servicio de Farmacia, Hospital Universitario de Fuenlabrada, 28942 Madrid, Spain; cristina.bravo@salud.madrid.org (C.B.); carolina.marino@salud.madrid.org (C.M.); 6Sistemas, Hospital Universitario de Fuenlabrada, 28942 Madrid, Spain; luisantonio.lechuga@salud.madrid.org (L.A.L.); jorge.tarancon@salud.madrid.org (J.T.); 7Servicio de Enfermedades Infecciosas, Hospital U. Ramón y Cajal, IRYCIS, 28034 Madrid, Spain

**Keywords:** COVID-19, SARS-CoV-2, waves, mortality

## Abstract

(1) Background: COVID-19 has evolved during seven epidemic waves in Spain. Our objective was to describe changes in mortality and severity in our hospitalized patients. (2) Method: This study employed a descriptive, retrospective approach for COVID-19 patients admitted to the Hospital de Fuenlabrada (Madrid, Spain) until 31 December 2022. (3) Results: A total of 5510 admissions for COVID-19 were recorded. The first wave accounted for 1823 (33%) admissions and exhibited the highest proportion of severe patients: 65% with bilateral pneumonia and 83% with oxygen saturation under 94% during admission and elevated levels of CRP, IL-6, and D-dimer. In contrast, the seventh wave had the highest median age (79 years) and comorbidity (Charlson: 2.7), while only 3% of patients had bilateral pneumonia and 3% required intubation. The overall mortality rate was 10.3%. The first wave represented 39% of the total. The variables related to mortality were age (OR: 1.08, 1.07–1.09), cancer (OR: 1.99, 1.53–2.60), dementia (OR: 1.82, 1.20–2.75), the Charlson index (1.38, 1.31–1.47), the need for high-flow oxygen (OR: 6.10, 4.94–7.52), mechanical ventilation (OR: 11.554, 6.996–19.080), and CRP (OR: 1.04, 1.03–1.06). (4) Conclusions: The variables associated with mortality included age, comorbidity, respiratory failure, and inflammation. Differences in the baseline characteristics of admitted patients explained the differences in mortality in each wave. Differences observed between patients admitted in the latest wave and the earlier ones suggest that COVID-19 has evolved into a distinct disease, requiring a distinct approach.

## 1. Introduction

On 31 December 2019, Zhu et al. reported a cluster of cases of ‘viral pneumonia’ in Wuhan, People’s Republic of China [1]. Subsequently, researchers labeled this virus SARS-CoV-2, and the World Health Organization termed the resulting illness COVID-19 [2]. The virus progressively spread worldwide, but factors such as virus variants, vaccination campaigns, behavioral measures, hospital capacity, and therapeutic advancements have been proposed to influence the evolution of the pandemic [3,4,5,6,7,8]. In any case, COVID-19’s incidence, hospitalization rates, and mortality showed significant variation across different epidemic waves, countries, and territories [9,10,11,12,13,14,15,16,17,18,19,20,21,22].

As of 14 June 2023, Spain held the 13th position globally for reported cases (13,890,555) and ranked 15th in the number of deaths (121,416) [23]. The country experienced seven distinct epidemic waves [24], as reported in other countries [25]. By 3 January, 2023, the Fuenlabrada region had confirmed 11,547 cases [26]. 

Some studies have investigated the clinical characteristics and outcomes of patients hospitalized for COVID-19 during the early waves [11,12,27,28,29,30]. Recent research has made comparisons in the waves attributed to the early Omicron lineages in the initial months of 2022, revealing significant differences from the earlier ones [10,25,31,32,33,34]. Subsequently, researchers identified multiple lineages arising from BA.2, BA.4, and BA.5, as well as recombinant lineages. These lineages gradually replaced each other, capitalizing on evolutionary advantages. Consequently, BA.5 gradually claimed global predominance in June 2022, and eventually global predominance was claimed by BQ.1* from November 2022 onwards, ultimately reigning as the predominant lineage at the end of our study [35]. To the best of our knowledge, no study has comprehensively compared all these waves together, including the last months of 2022. Therefore, having this information available in various countries is crucial to prepare for future epidemics.

Our main objective was to describe the characteristics of our hospitalized COVID-19 patients, conducting comparisons across seven epidemic waves and evaluating the virus’s impact on mortality and severity throughout the pandemic.

## 2. Materials and Methods

### 2.1. Study Design and Setting

This was retrospective, cross-sectional descriptive research, from 1 March 2020 to 31 December 2022, carried out at the Hospital Universitario de Fuenlabrada. 

This second-level hospital serves a population of approximately 220,000 people in the south of Madrid. Since its opening in 2004, it has been the referral hospital for patient admissions in this area, including all those diagnosed with COVID-19.

First objective: describe the characteristics of hospitalized COVID-19 patients.

Second objective: identify potential risk factors associated with mortality and severity in COVID-19 patients.

### 2.2. Subjects

Inclusion criterion: All consecutive adult patients (over 16 years old) admitted to the Hospital Universitario de Fuenlabrada with a diagnosis of COVID-19 during the study period.

Exclusion criteria:Asymptomatic patients, regardless of the results of microbiological tests.Patients with symptoms not considered compatible with COVID-19, regardless of the results of microbiological tests.

The availability of tests varied during the study period, which influenced the definition of COVID-19:From March 2020 to June 2020, microbiological tests were not available for most patients. We classified cases as COVID-19 when compatible clinical symptoms were present, if clinicians did not diagnose any other infectious disease.From June 2020 to April 2022, the hospital implemented a comprehensive strategy to prevent nosocomial transmission by testing all admitted patients for the SARS-CoV-2 antigen or nucleic acid detection. We classified cases as COVID-19 when both a positive microbiological test result and a clinical diagnosis were present.From April 2022 to December 2022, the hospital only tested for the SARS-CoV-2 antigen or nucleic acid detection in admitted patients with a clinical suspicion of COVID-19. We only classified cases as COVID-19 when they had a positive microbiological test.

### 2.3. Definition of Epidemic Waves

In our country, dates for seven epidemic waves had been proposed, based on variations in the 14-day cumulative incidence [24]. For our analysis, we decided to adapt the wave dates by considering the increase in the number of weekly COVID-19 admissions as the turning point between waves, instead of relying solely on the 14-day incidence. We believe this approach better reflects the hospital’s impact, as factors like the number of tests conducted and the 14-day reporting delay can influence the incidence. We established this definition before conducting the descriptive study and statistical analyses.

### 2.4. Variables

Complete definitions are available in the Appendix A.

The main outcome variables were the need for mechanical ventilation upon admission to the hospital and any-cause mortality at 3 months after admission. The following variables were recorded: admission date (defining the COVID-19 wave); sociodemographic variables (age, gender, and place of birth); comorbidities, including hypertension, diabetes, cardiopathy, chronic obstructive pulmonary disease (COPD), asthma, oncological disease, HIV, dementia, and the Charlson comorbidity index; vaccination status at the time of admission; disease severity, determined via chest X-ray at admission (normal/unilateral/bilateral pneumonia), oxygen saturation in the emergency department and throughout admission (absolute value and categorical (less than 94%)), maximum oxygen support required during admission (no oxygen/low oxygen flow/high oxygen flow/intubation), and length of hospitalization and ICU stay; levels of C-reactive protein (CRP), interleukin 6 (IL-6), D-dimer, and ferritin; and the use of any of these medications: remdesivir, tocilizumab, baricitinib, corticosteroids, and prophylactic low-molecular-weight heparin (LMWH) [14]. 

### 2.5. Statistics

Considering the admission date, the enrolled patients were classified into seven COVID-19 wave groups. Categorical variables are presented as numbers, proportions, and 95% confidence intervals. Quantitative variables were assessed for normal distributions using the Kolmogorov–Smirnov test. Normally distributed variables are presented as means and standard deviations, and non-normally distributed variables are presented as medians and interquartile ranges. 

A comparison of the clinical characteristics among the seven COVID-19 wave groups was conducted. Categorical variables were compared using the chi-square test or Fisher’s exact test (if any observed frequency was less than 5 or 20%). For normally distributed variables, an ANOVA was employed, while for non-normally distributed variables, the Kruskal–Wallis test was used. When significant differences were identified, the Tukey–Kramer method was applied to investigate distinctions within the respective waves. A significance level of *p* < 0.05 was set to determine statistical significance.

A univariate analysis was conducted for each variable in relation to the dependent variables “overall mortality at 3 months” and “need for mechanical ventilation”. Variables that exhibited a significance level of *p* < 0.10 in the univariate analyses were subsequently incorporated into a logistic regression analysis to identify potential risk factors for the aforementioned dependent variables. Two final models are shown. The first model excludes the variable “maximum IL-6 value,” due to the noteworthy prevalence of missing cases associated with this specific variable, because IL-6 was not assessed in 2673 subjects. The second model includes this variable because of its relevance. Associations were expressed as adjusted ORs and 95% CIs.

In the first model of mortality, the following variables were in the final analyses due to values of *p* < 0.1 in the univariate analysis: COVID-19 wave; vaccination status at the time of admission; age; immigrant status; Charlson comorbidity index; hypertension; diabetes; cardiopathy; asthma; COPD; oncological disease; dementia; oxygen saturation; high O_2_ flow; mechanical ventilation; bilateral infiltrates on X-ray; levels of C-reactive protein (CRP), D-dimer, and ferritin; tocilizumab; corticosteroids; and LMWH. In the second model, “maximum IL-6” was included.

In first model of mechanical ventilation, the following variables were in the final analyses due to values of *p* < 0.1 in the univariate analysis: COVID-19 wave; vaccination status at the time of admission; gender; age; immigrant status; Charlson comorbidity index; hypertension; diabetes; cardiopathy; oncological disease; dementia; oxygen saturation; bilateral infiltrates on X-ray; levels of C-reactive protein (CRP), D-dimer, and ferritin; tocilizumab; corticosteroids; and LMWH. In the second model, “maximum IL-6” was included.

All analyses were performed using IBM SPSS Statistics for Windows, Version 25.0 (IBM Corp., Armonk, NY, USA).

## 3. Results

### 3.1. Duration of the Epidemic Waves

Based on the increase in the number of weekly admissions, we defined the following dates for each epidemic wave (Figure 1):

First wave: 4 March 2020 to 2 July 2020, with a peak on 31 March 2020.Second wave: 15 July 2020 to 25 November 2020, with a peak on 25 September 2020.Third wave: 26 November 2020 to 28 February 2021, with a peak on 25 January 2021.Fourth wave: 1 March 2021 to 30 June 2021, with a peak on 16 April 2021.Fifth wave: 1 July 2021 to 30 September 2021, with a peak on 23 August 2021.Sixth wave: 1 October 2021 to 4 April 2022, with a peak on 17 January 2022.Seventh wave: 5 April 2022 to 31 December 2022, with a peak on 28 June 2022.

### 3.2. Description

There were 5510 COVID admissions, corresponding to 5001 patients, and 509 admissions were second episodes (9%) (Table 1). Nearly 50% of the total admissions occurred in the first two waves, while hospitalizations decreased in subsequent waves.

Table 2 presents the baseline characteristics of the patients by epidemic wave. The median age and the burden of comorbidity were significantly higher in the last two waves. The highest proportions of immigrant patients occurred in the second and fifth waves. Only 14 people living with HIV (PLHIV) required admission: 2 needed mechanical ventilation and 1 died.

We show the clinical variables in Table 3. The highest proportion of patients in the second wave had better oxygen saturation at admission. Patients from the first and third waves experienced a worsening of their respiratory condition during hospitalization. Patients from the first wave had the highest proportion of bilateral pneumonia. This was associated with a higher need for high-flow oxygen in this wave but not for mechanical ventilation. Bilateral pneumonia was very uncommon in the seventh wave, and this resulted in reduced use of oxygen requirements. Inflammatory parameters were higher in the first wave, but corticosteroids, tocilizumab, and baricitinib were less used in this wave. During the seventh wave, remdesivir was used more frequently, while heparin was employed less.

Regarding patients with a maximum CRP greater than 7.5 mg/dL, 73% received steroids and 36% received tocilizumab. For patients with IL-6 levels greater than 40 pg/mL before treatment, 95% received steroids and 100% received tocilizumab.

The median length of stay for patients was 7.1 days, which was lower in the last wave. Of the 5510 admissions, 358 required mechanical ventilation (6.5% of the total admissions, 7% of patients), and 514 patients (10.3%) died within 3 months of admission. The lowest mortality occurred in the fourth and fifth waves. The first wave accounted for 39% of all deaths.

### 3.3. Factors Associated with COVID-19 Mechanical Ventilation

Table 4 shows the results of the multivariate analysis of mechanical ventilation. When including the maximum IL-6 value, the same variables were retained, along with the IL-6 level (continuous) (OR: 1.001 (95% CI: 1.000–1.001) per 1.0 pg/mL, *p* < 0.001).

### 3.4. Factors Associated with COVID-19 Mortality

Table 5 shows the results of the multivariate analysis of 3-month mortality. When including the maximum IL-6 value, the same variables were maintained, except for CRP and oxygen saturation at admission and the inclusion of IL-6 as a continuous variable (OR: 1.001 (95% CI: 1.000–1.001) per 1.0 pg/mL, *p* = 0.002).

## 4. Discussion

We provide an overview of COVID-19 hospitalizations over almost three years, divided into distinct waves, within a single institution. There are limited studies that compare the progression of COVID-19 patients over such an extended period [10,32]. It gives a global view of the pandemic that is worth considering. 

It is widely proposed that factors such as virus variants, vaccination campaigns, behavioral measures, hospital capacity, and therapeutic advancements have contributed in complex ways to the evolution of COVID-19 in hospitals [3,4,5,6,7,8]. Other factors, such as the availability of PCR testing or changes in admission and discharge criteria, may have influenced the measurement of the disease’s impact [6,10]. These factors have been highly variable from one country to another and even within the same country and have significantly affected the differences in the numbers and characteristics of epidemic waves in each geographical area [10,11,12,13,14,15,16,17,18,19,20,21,22]. Here, we discuss some hypotheses, and we can only establish the association, but not causation, between certain events and peaks of hospitalizations in our geographic area.

We observed that the first wave had the greatest numbers of hospitalized patients and deaths. This wave also included the most severely affected patients, marked by a deterioration in their respiratory condition, a higher incidence of bilateral pneumonia, and elevated levels of inflammatory markers. Similarly affected regions during the pandemic’s early stages also reported alarming data in the initial wave [22,27,29]. Notably, this severity was not associated with a high percentage of patients requiring mechanical ventilation (6%) compared to the other waves. Similar series also demonstrated low intubation rates in this wave, ranging between 8 and 12% [27,28,36,37,38]. In other regions where the initial wave was milder, a higher intubation rate (20%) was reported [10,21]. We can speculate that, in the most strained hospital, not all patients requiring intubation received it during this wave due to resource limitations. In line with our results, other hospitals utilized fewer anti-inflammatory treatments (corticosteroids, tocilizumab, and baricitinib) compared to later waves [10,32]. It is likely that the lack of evidence influenced the scarce use of these treatments [39]. Nevertheless, our rates of corticosteroid, tocilizumab, and prophylactic LMWH use were greater when compared to the other large series [32,36,38].

Most studies that have compared the second wave with the first found that the second one was milder in terms of hospitalizations and severity [10,12,13,22,27]. In line with our findings, previous reports have indicated that COVID-19 during this wave was characterized by a younger age and a higher proportion of immigrants [27]. These data contrast with those observed in the third wave. Older patients experiencing a worsening of their respiratory condition during hospitalization and a higher utilization of anti-inflammatory treatments characterized this wave. We can assume, in our patients, that better use of treatments may partially explain why the inflammatory parameters did not reach the levels of the first wave. In the UK, the increased severity of this wave was attributed to the emergence of the Alpha variant during winter 2021 [40], but this variant entered our country at a later time. In Japan or Congo, this wave, also during winter 2021, also had a higher percentage of patients and deaths [10,16]. In other countries, this wave did not even occur [18,20]. 

Starting in early 2021, the B.1.1.7 (Alpha, UK variant) replaced the original strain in our region [35]. This strain dominated during the fourth wave, similar to the situation reported in Japan [10]. In both countries, it was quickly replaced by B.1.617.2 (Delta, Indian variant) during the fifth wave [10,35]. The Delta variant was considered even more severe than Alpha and struck the healthcare systems in countries such as the UK, India, Bangladesh, Japan, and Argentina [3,10,14,18,20]. In this way, we found in these fourth and fifth waves the highest proportion of patients requiring intubation. However, this also coincided with the onset and progression of vaccination campaigns [25,41]. Vaccination efforts are considered a significant factor in reducing hospitalization and mortality among patients with COVID-19 [4,7,33,42,43,44]. It has been reported that the early mass vaccination of people over 60 years of age prevented hospitalizations during the spring of 2021 in Spain [8]. This could explain why hospitalized patients were younger in countries with a high percentage of vaccinated individuals. Both factors, vaccination and a younger age, could have led to lower mortality rates in both these waves, like in our hospital [10,22,32]. 

Throughout 2022, the Omicron lineages successively gained prominence. BA.1 emerged in late December 2021 and was probably the trigger for the sixth wave. Subsequently, researchers identified multiple lineages arising from BA.2, BA.4, and BA.5, as well as recombinant lineages. These lineages gradually replaced each other, capitalizing on evolutionary advantages. Consequently, BA.2 gradually claimed global predominance from March to June 2022 [35]. In our sixth wave, age and comorbidity were higher than in previous waves, according to similar studies [19,25,33,43,45]. Respiratory data appeared to improve, with fewer pneumonias and better oxygen saturation than in the first and third waves. However, mortality data worsened compared to the fourth and fifth waves. This is striking when compared to most countries, where the overall introduction of Omicron led to decreases in severity and mortality [19,22,33,43,44,45,46,47,48]. This could be consistent with the increased severity of the previous Delta wave in these regions.

In June 2022, BA.5 emerged, followed by BQ.1* in November 2022, eventually becoming the predominant lineage by the end of our study [35]. There are no relevant reports on the pandemic’s behavior in the latter months of 2022, so our data are very relevant. Our seventh wave had a very low incidence of bilateral pneumonia, leading to decreases in oxygen requirements and mechanical ventilation. It is important to remember that we excluded asymptomatic patients and those with non-definitive COVID symptoms. The median age and comorbidity burden were the highest. There was an increased use of remdesivir and a decreased use of heparin. The median patient stay was low (5.8 days), but mortality remained statistically similar to the first waves. These data underscore the paradigm shift needed to address COVID-19. Our findings indicate that COVID-19 is no longer a severe disease, but it continues to cause mortality in vulnerable populations. Patients in the seventh wave were old, with comorbidities, and developed a non-severe respiratory disease. Underlying conditions were responsible for mortality rather than COVID-19 itself. From now on, COVID-19 strategies should focus on nosocomial prevention in vulnerable patients [49], and treatments should probably target the virus more than inflammation [50].

Only 14 PLHIV needed hospitalization, with only one death. The largest study on PLHIV and COVID-19 did not indicate any elevated risk or increased mortality among hospitalized patients in this group [51].

The variables associated with mortality were not surprising: age, cancer, dementia, Charlson score, need for high-flow oxygen or mechanical ventilation, and inflammatory markers. These factors had been described in previous studies [29,31,52,53,54,55,56]. We show that age and comorbidity played a more significant role in recent waves, in which the respiratory and inflammatory status of COVID-19 patients has improved. On the other hand, the use of prophylactic LMWH was associated with lower mortality. This treatment is now strongly recommended in all hospitalized COVID-19 patients [57]. As a limiting factor, we could not distinguish patients without prophylactic LMWH from those who received anticoagulated doses before or during admission, variables that are likely associated with different pre-existing comorbidities and COVID-19 severity. We chose the 3-month mortality, as it corresponds to the definition proposed for long COVID [58]. In this way, we aimed to assess the mortality of “short COVID”. A limitation is that we lack the cause of death, which is particularly relevant in cases of death following discharge.

The multivariate analyses did not reveal any specific wave associated with either mortality or intubation. This suggests that the differences are unlikely to be caused by unexamined variables. However, in our study is not possible to rule out this possibility. We only found one study conducted in Congo that performed a similar analysis, identifying an association between mortality and the wave in winter of 2021 [16]. 

Finally, it is also noteworthy that we did not find an association between vaccination and overall mortality. In this regard, we previously assessed the impact of vaccination in the fourth and fifth waves in our center [42]. In summary, although the mortality rate was higher in the vaccinated group in terms of percentage, this was due to a higher accumulation of comorbidity in this group, and we can ascertain that it would have been even higher without the vaccine [4,8].

The main limitations of our study include its retrospective design. We tried to overcome these limitations with a rigorous method. We included all COVID-19 patients admitted to the hospital, accounting for a large number of patients and the quality of the variables collected prospectively in an electronic database for subsequent analysis. Measuring the data at a single center may limit the generalizability, but it adds homogeneity to the results. The variability in admission criteria and COVID-19 definitions across different hospitals makes it challenging to extend clinical data, especially mortality rates. Most SARS-CoV-2-infected patients do not require hospitalization, so we cannot report the overall mortality of SARS-CoV-2 infections. 

## 5. Conclusions

In this single-center study, we observed significant changes in the characteristics and evolution of hospitalized COVID-19 patients. Variables associated with mortality included age, comorbidity, respiratory failure, and inflammation. However, the mortality and intubation rates did not show an independent association with the COVID-19 wave. This suggests that the differences observed between waves in both variables in our series were mainly due to the different baseline characteristics of the patients admitted during each wave.

Disparities between patients admitted in the latest wave and the earliest imply that COVID-19 has evolved into a distinct disease. Currently, COVID-19 is a mild disease that affects older patients with a high burden of morbidity. However, it leads to high mortality due to the underlying characteristics of these patients. Hospitals and healthcare services must adapt to this new situation, requiring a distinct approach.

## Figures and Tables

**Figure 1 viruses-15-01839-f001:**
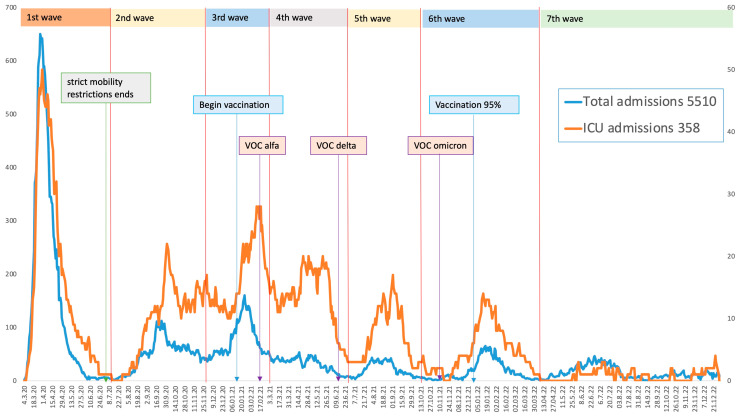
Distribution of hospital admissions and ICU admissions during the study period across different waves.

**Table 1 viruses-15-01839-t001:** Total admissions and patients by epidemic wave *.

Wave	First	Second	Third	Fourth	Fifth	Sixth	Seventh	Total
Patients	1735 (35%)	900 (18%)	823 (17%)	414 (8%)	291 (6%)	441 (8%)	397 (8%)	5001
Total admissions	1823 (33%)	980 (18%)	900 (16%)	472 (9%)	322 (6%)	522 (9%)	491 (9%)	5510
Second episodes	88 (5%)	80 (8%)	77 (9%)	58 (12%)	31 (10%)	81 (16%)	94 (19%)	509 (9%)

* All percentages refer to the total number of patients and admissions, except in the case of second episodes, in which percentages refer to the number of admissions in each wave.

**Table 2 viruses-15-01839-t002:** Baseline characteristics by epidemic wave.

Wave	First	Second	Third	Fourth	Fifth	Sixth	Seventh	Total	*p*
Patients	1735 (35%)	900 (18%)	823 (17%)	414 (8%)	291 (6%)	441 (8%)	397 (8%)	5001	
Male sex	957 (55%)	464 (52%)	472 (57%)	232 (56%)	163 (52%)	228 (52%)	200 (50%)	2743 (54%)	0.073
Age ^2^	64 (54–74)	60 (48–72)	65 (54–76)	61 (50–72)	53 (38–68)	68 (57–78)	79 (71–87)	65 (53–76)	<0.001 ^3^
Place of birth									
Spain ^1^	1435 (84, 82–86)	600 (68, 67–71)	700 (87, 84–90)	330 (80, 76–84)	188 (66, 61–71)	368 (85, 82–88)	382 (96, 94–98)	4003 (81, 80–82)	<0.001
Latin America	168 (10%)	156 (17%)	63 (8%)	49 (12%)	30 (11%)	24 (6%)	6 (2%)	495 (10%)	<0.001
North Africa	26 (2%)	66 (7%)	16 (2%)	14 (3%)	25 (8%)	12 (3%)	3 (1%)	162 (3%)	<0.001
Vaccinated	0	0	1 (0.1%)	20 (5%)	112 (39%)	328 (74%)	159 (88%)	620 (13%)	<0.001
Comorbidities									
Charlson index ^4^	1.3 (2.3)	1.4 (2.3)	1.5 (2.3)	1.4 (2.5)	1.4 (2.4)	2.2 (2.8)	2.7 (2.7)	1.5 (2.4)	<0.001 ^5^
Hypertension	816 (47%)	371 (41%)	425 (52%)	185 (45%)	114 (39%)	253 (57%)	150 (68%)	2314 (48%)	<0.001
Diabetes	221 (13%)	96 (11%)	113 (14%)	33 (8%)	26 (9%)	58 (13%)	48 (22%)	595 (12%)	<0.001
Cardiopathy	77 (4%)	42 (5%)	42 (5%)	13 (3%)	19 (7%)	21 (5%)	24 (11%)	238 (5%)	0.002
COPD ^6^	171 (10%)	72 (8%)	75 (9%)	32 (8%)	23 (7%)	62 (14%)	88 (35%)	523 (11%)	<0.001
Asthma	158 (9%)	66 (7%)	78 (10%)	31 (8%)	24 (8%)	44 (10%)	30 (13%)	431 (9%)	0.148
Cancer	323 (19%)	155 (17%)	149 (18%)	57 (14%)	40 (14%)	119 (27%)	109 (45%)	952 (20%)	<0.001
Dementia	51 (3%)	30 (3%)	27 (3%)	12 (3%)	10 (3%)	34 (8%)	52 (21%)	216 (5%)	<0.001
PLHIV ^7^	5	2	1	2	0	1	3	14 (0.3%)	0.228

^1^ n (%, 95% CI). ^2^ Years: median (interquartile range). ^3^ Tukey–Kramer for age (*p* < 0.05): all, except 1st vs. 3rd, 2nd vs. 4th, and 3rd vs. 6th. ^4^ Units: mean, standard deviation. ^5^ Tukey–Kramer for Charlson (*p* < 0.0001) (6th and 7th waves vs. each other). ^6^ COPD: chronic obstructive pulmonary disease. ^7^ PLHIV: people living with human immunodeficiency virus.

**Table 3 viruses-15-01839-t003:** Clinical variables by wave.

Wave	First	Second	Third	Fourth	Fifth	Sixth	Seventh	Total	*p*
Patients	1735 (35%)	900 (18%)	823 (17%)	414 (8%)	291 (6%)	441 (8%)	397 (8%)	5001	
Oxygen saturation on admission under 94% ^1^	815 (45, 43–47)	333 (34, 31–37)	392 (44, 41–47)	207 (44, 39–49)	129 (40, 34–46)	191 (37, 32–41)	123 (45, 40–50)	2190 (41, 40–42)	<0.001
Worst oxygen saturation under 94% ^1^	1512 (83, 81–85)	738 (75, 72–78)	759 (84, 81–87)	367 (78, 74–82)	237 (74, 70–78)	392 (75, 71–79)	223 (81, 77–85)	4228 (80, 79–81)	<0.001
Oxygen requirements									
None ^1^	415 (24, 22–26)	266 (30, 27–33)	165 (20, 17–23)	86 (21, 17–25)	50 (18, 14–22)	114 (26, 22–30)	56 (30, 25–35)	1152 (25, 24–26)	<0.001
Low oxygen flow ^1^	869 (51, 49–53)	434 (50, 47–53)	430 (53, 50–56)	214 (53, 48–58)	156 (55, 49–51)	234 (53, 49–58)	118 (64, 59–69)	2455 (52, 51–53)	<0.001
High oxygen flow ^1^	321 (19, 17–21)	107 (12, 10–14)	151 (19, 16–22)	53 (13, 10–16)	42 (15, 11–19)	64 (15, 12–18)	6 (3, 1–5)	744 (16, 15–17)	<0.001
Mechanical ventilation ^1^	93 (6, 5–7)	67 (8, 6–10)	64 (8, 6–10)	48 (12, 9–15)	34 (12, 8–16)	28 (6, 4–8)	5 (3, 1.5)	339 (7, 6–8)	<0.001
Bilateral infiltrates on chest X-ray ^1^	1162 (67, 66–69)	310 (33, 30–35)	433 (48, 45–51)	175 (40, 35–45)	145 (56, 50–62)	149 (29, 25–33)	9 (3, 7–11)	2383 (47, 46–48)	<0.001
CRP ^2,3^	109 (50–170)	87 (30–144)	93 (38–148)	99 (44–154)	85 (26–145)	72 (12–132)	59 (17–101)	94 (30–150)	<0.001 ^4^
IL-6 ^2,3^	44 (1–101)	34 (0–90)	50 (1–130)	47 (1–127)	42 (1–120)	27 (1–76)	12 (1–28)	40 (0–104)	<0.001 ^4^
DD ^2,3^	1021 (256–1785)	901 (232–1570)	1092 (123–2061)	986 (192–1779)	942 (272–1611)	1099 (319–1879)	943 (540–1345)	1015 (266–1764)	<0.001 ^4^
Ferritin ^2,3^	543 (94–991)	536 (155–917)	572 (144–1000)	610 (110–1110)	539 (174–943)	412 (72–762)	196 (16–416)	528 (121–935)	<0.001 ^4^
Remdesivir	0	15 (2%)	9 (1%)	2 (0.5%)	0	1 (0.2%)	9 (5%)	36 (1%)	<0.001
Corticosteroids ^1^	715 (41, 39–43)	609 (68, 65–71)	679 (83, 80–86)	324 (78, 74–82)	236 (81, 76–86)	337 (76, 72–80)	123 (69, 64–74)	3023 (63, 62–64)	<0.001
Tocilizumab ^1^	257 (15, 13–17)	261 (29, 26–32)	347 (42, 39–45)	163 (39, 34–44)	102 (35, 30–40)	100 (23, 19–27)	9 (5, 3–7)	1239 (26, 25–27)	<0.001
Baricitinib ^1^	17 (1, 0–2)	5 (15, 13–17)	6 (1, 0–2)	10 (2, 1–3)	46 (16, 12–20)	57 (13, 10–16)	3 (2, 1–3)	144 (3, 2–4)	<0.001
pLMWH ^1,5^	1450 (84, 82–86)	801 (89, 87–91)	750 (91, 89–93)	398 (87, 84–90)	284 (89, 85–93)	365 (83, 79–87)	121 (68, 63–73)	4103 (86, 85–87)	<0.001
Total length of stay, days ^2^	7.8 (4.3–11.3)	7.0 (3.5–10.5)	7.2 (3.2–11.2)	6.9 (2.4–11.4)	7.0 (3.0–11.0)	5.8 (2.3–9.3)	5.8 (3.3–7.3)	7.1 (3.1–11.1)	<0.001 ^6^
Deaths ^1^	200 (11.5, 10.0–13.0)	89 (9.9, 8.0–11.8)	93 (11.3, 9.1–13.5)	29 (7, 4.5–9.5)	19 (6.5, 3.7–9.3)	53 (12, 9–15)	31 (8, 5.3–10.7)	514 (10.3, 9.5–11.1)	0.040

^1^ n (%, 95% CI). ^2^ Median (interquartile range). ^3^ C-reactive protein (highest level), mg/L; interleukin 6 (highest level), pg/mL; D-dimer (highest level), ng/mL; ferritin (highest level), ng/mL. ^4^ Tukey–Kramer CRP (*p* < 0.05): 1st vs. 2nd, 3rd, 5th, 6th, 7th; 6th vs. 4th, 5th; 7th vs. 1st, 2nd, 3rd, 4th, 5th. Tukey–Kramer IL6 (*p* < 0.05): 1st vs. 2nd, 3rd, 6th, 7th. Tukey–Kramer DD (*p* < 0.05): 1st vs. 2nd. Tukey–Kramer ferritin (*p* < 0.05): 7th vs. 1st, 2nd, 3rd, 4th, 5th. ^5^ Prophylactic low-molecular-weight heparin. ^6^ Tukey–Kramer stay (*p* < 0.05): 7th vs. 1st, 3rd, 4th.

**Table 4 viruses-15-01839-t004:** Multivariate analysis of mechanical ventilation.

Predictive Variables Included in the Model	OR (95% CI)	*p*
Cancer (categorical)	0.49 (0.30–0.81)	0.046
Worst oxygen saturation < 94% (categorical)	7.36 (2.04–26.61)	<0.001
Bilateral infiltrates (categorical)	4.03 (3.27–4.95)	<0.001

**Table 5 viruses-15-01839-t005:** Multivariate analysis of 3-month mortality.

Variables	OR (95% CI)	*p*
Age (continuous, per 1.0 year)	1.08 (1.07–1.09)	<0.001
Charlson index (continuous, per 1.0)	1.38 (1.31–1.47)	<0.001
Cancer (categorical)	1.99 (1.53–2.60)	<0.001
Dementia (categorical)	1.82 (1.20–2.75)	0.010
High O_2_ flow (categorical)	10.243 (6.880–15.251)	<0.001
Mechanical ventilation (categorical)	11.554 (6.996–19.080)	<0.001
C-reactive protein (continuous, per 1.0 mg/dL)	1.04 (1.03–1.06)	<0.001
Low-molecular-weight heparin (categorical)	0.41 (0.30–0.57)	<0.001

## Data Availability

Database belongs to the Hospital Universitario de Fuenlabrada. Restrictions apply to the availability of these data. Data were obtained from patients hospitalized and are available [juanvictor.san@salud.madrid.org] with the permission of Gerencia de Hospital Universitario de Fuenlabrada.

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
