# Peer review of "Seven Epidemic Waves of COVID-19 in a Hospital in Madrid: Analysis of Severity and Associated Factors"

_viruses, 2023, doi:10.3390/v15091839_

Round 1
Reviewer 1 Report
General comments
The manuscript by San Mart´ın et al. presents a retrospective analysis of COVID waves in the Hospital de Fuenlabrada in Madrid up until the end of 2022. There is not much wrong with the manuscript, which is fairly straightforward overall, and it can be published with some corrections and clarifications.
Major comments
1. The B.1.1.529/Omicron lineage is treated as a single variant, but in reality “Omicron” waves consisted of quite different antigenically (and also with some differences elsewhere) variants that were never given proper Greek letters only because of the political mandate to declare the pandemic over. Had it not been for that factor, we would have exhausted the Greek alphabet by now if we followed the previously established criteria. The scientifically correct thing here is to properly distinguish the different variants driving Omicron waves (BA.1/BA.2/BA.5/XBB.*/etc.)
2. Table 6 (and some others, but this one in particular) could benefit from adding confidence intervals on the proportions.
Minor comments
1. Lines 44–45: the pandemic wasn’t really declared “over” (nor is it over in reality, of course); what was declared over was the global public health emergency associated with it.
2. Lines 46–47: like most other countries, Spain did not test and report all cases and deaths. Excess mortality is the proper measures of the impact of the pandemic, and in Spain’s case excess deaths are at least 20% higher than the official ones
3. Lines 91–92: “positive SARS-CoV-2 antigen test or molecular detection” implies that antigen tests are somehow not “molecular” tests, which of course they are. It is better to just say “PCR” or “nucleic acid” instead.
4. Lines 93–94: if testing is said to have been universal until April 2022, what does that mean for the period between April 2022 and December 2022? How many cases in the hospital might have been missed as a result of changes in policies?
5. Line 224: “2,673 subject” probably should be “2,673 subjects”
6. Lines 351–352: “Main limitations of our study include the retrospective design and being carried out at a single” – this sentence appears to be incomplete.
none
Author Response
Dear reviewer,
We sincerely appreciate your interest in our manuscript. We have carefully addressed your inquiries and incorporated the suggested revisions. In response to feedback from other reviewers, we have made significant changes to the introduction, discussion, and references. We apologize for any discrepancies and trust that you will find the updated manuscript even more engaging.
Best regards, the authors
General comments
Major comments
- The B.1.1.529/Omicron lineage is treated as a single variant, but in reality “Omicron” waves consisted of quite different antigenically (and also with some differences elsewhere) variants that were never given proper Greek letters only because of the political mandate to declare the pandemic over. Had it not been for that factor, we would have exhausted the Greek alphabet by now if we followed the previously established criteria. The scientifically correct thing here is to properly distinguish the different variants driving Omicron waves (BA.1/BA.2/BA.5/XBB.*/etc.)
Answer: Thank you for this interesting insight. We have added a note in the manuscript to clarify this point, explaining how multiple Omicron lineages caused the last waves. Nonetheless, we did not find any significant clinical impact, and this is of particular interest.
- Table 6 (and some others, but this one in particular) could benefit from adding confidence intervals on the proportions.
Answer: We added CI in tables
Minor comments
- Lines 44–45: the pandemic wasn’t really declared “over” (nor is it over in reality, of course); what was declared over was the global public health emergency associated with it.
Answer: we erased this sentence
- Lines 46–47: like most other countries, Spain did not test and report all cases and deaths. Excess mortality is the proper measures of the impact of the pandemic, and in Spain’s case excess deaths are at least 20% higher than the official ones
Answer: we totally agree, but is impossible to be sure about excess deaths in other countries too. We aim to provide the official figures to contextualize our hospital.
- Lines 91–92: “positive SARS-CoV-2 antigen test or molecular detection” implies that antigen tests are somehow not “molecular” tests, which of course they are. It is better to just say “PCR” or “nucleic acid” instead.
Answer: thank you, corrected
- Lines 93–94: if testing is said to have been universal until April 2022, what does that mean for the period between April 2022 and December 2022? How many cases in the hospital might have been missed as a result of changes in policies?
Answer: We try to clarify. The hospital changed strategy to prevent nosocomial transmission and stopped universal test as part of prevention of nosocomial SARS-CoV-2 transmission. Instead, we adopted extreme precaution with vulnerable subjects (individual rooms and standard isolation measures). No cases of COVID were missed, because microbiological tests were done in all symptomatic cases. We never considered asymptomatic infected as COVID.
- Line 224: “2,673 subject” probably should be “2,673 subjects”
Answer: erased because of suggestion of another reviewer
- Lines 351–352: “Main limitations of our study include the retrospective design and being carried out at a single” – this sentence appears to be incomplete.
Answer: corrected
Comments on the Quality of English Language
None

Reviewer 2 Report
I really appreciate the work done by the authors: the paper is very synthetic, it is focused on some aspects that can be useful for future comparison, it make the analysis of a practical case study.
My personal opinion is strongly affected by the novelty of the paper: is it enough the content argued ?
I don't know. The contents are correct but my personal question is strongly related if it can be useful for the Scientific Community, it it will be cited,
I suggest to stress the text with other experiences, to compare the data collected with other data from international colleagues, to improve the text.. also considering how the quality of the spaces, the building typology of the hospital can facilitate the management of the facility, etc. (in that case please refer to the studies by Stefano Capolongo and the recent technical brief published by WHO in 2023 about the hospital of the future).
The text has a good sound. It works well.
no mistakes has been selected.
Author Response
Dear Reviewer:
We sincerely appreciate your interest in our manuscript. We have carefully addressed your inquiries and incorporated the suggested revisions. In response to feedback from other reviewers, we have made significant changes to the introduction, discussion, and references. We apologize for any discrepancies and trust that you will find the updated manuscript even more engaging.
Best regards, the authors
REVIEWER 2
Suggestions for Authors
I really appreciate the work done by the authors: the paper is very synthetic, it is focused on some aspects that can be useful for future comparison, it make the analysis of a practical case study.
My personal opinion is strongly affected by the novelty of the paper: is it enough the content argued ?
I don't know. The contents are correct but my personal question is strongly related if it can be useful for the Scientific Community, it it will be cited,
I suggest to stress the text with other experiences, to compare the data collected with other data from international colleagues, to improve the text.. also considering how the quality of the spaces, the building typology of the hospital can facilitate the management of the facility, etc. (in that case please refer to the studies by Stefano Capolongo and the recent technical brief published by WHO in 2023 about the hospital of the future).
Response: We have revised the introduction and discussion sections, emphasizing our strengths, particularly the novelty of our paper. Additionally, we have incorporated numerous comparisons with other studies, including reference to the WHO document on the hospital of the future.
Comments on the Quality of English Language
The text has a good sound. It works well.
no mistakes has been selected

Reviewer 3 Report
In my opinion, this is an interesting, timely, and necessary manuscript. The study describes and documents the COVID-19 pandemic in the years 2020 and 2021 using a hospitalization database from a local area in Spain. It analyzes severity indicators such as hospital admissions, ICU admissions, and deaths. These indicators are important in pandemic situations and are of great importance for vaccine effectiveness studies. But more importantly, it assesses the impact of the pandemic in terms of hospitalization. Studies like this allow epidemiologists to complement and improve their population databases, although they represent a subset of the information generated at the population level. And they are also useful for monitoring pandemic situations like COVID-19.
However, I have several concerns about it.
GENERAL
Overall, although English is not my native language, the manuscript requires intensive English editing. Some sentences are difficult to understand, and certain connectors or conjunctive words are not used properly or are limited, resulting in poor English. Additionally, there are sentences that are too long, spanning 3 to 4 lines. I highly recommend seeking professional English editing services.
One notable issue is that the objectives are not clearly stated. In the abstract, it merely states: "Our objective was to study mortality and severity, and associated factors in our hospitalized patients." However, it does not specify the context or focus of this study. For instance, it does not clarify whether the objective is to study mortality and severity in terms of ICU admissions or the need for oxygen support. While reading the manuscript, I formed an overall idea of the objectives, and they seemed different from what was stated in both the abstract and the Introduction section.
After thoroughly reading the manuscript, and considering it is just my opinion, the first objective appears to be describing the burden of COVID-19 in the author's institution. On the other hand, the second objective seems to be highlighting the association of several variables with mortality and/or severity. Therefore, the proposed study should be categorized as a retrospective, cross-sectional descriptive research. However, I will emphasize later the type of analyses required, as it seems that the authors are not clear on the difference between a cross-sectional study and a longitudinal study. This is evident from their reporting of HR (hazard ratios) instead of OR (odds ratios).
INTRODUCTION
The first paragraph should serve as an effective introduction to the main topic, COVID-19. However, the current version is tangled up and chaotic due to excessively long sentences and the use of web page references instead of scientific papers. Therefore, we propose breaking the first paragraph into two or three, focusing on presenting the topic clearly and concisely.
The authors used too long sentences, such as “On March 11, 2020, as the virus progressively spread worldwide, the WHO declared it a pandemic, which on May 05, 2023, was declared over.” Also, this first paragraph cited eight references from web pages. The first eight references should not be web pages, but relevant scientific papers.
Paragraph 2 (Factors Influencing Waves): In the subsequent section (line 50 to line 66), the authors enumerate several disjointed topics and ideas related to the emergence of COVID-19 waves. However, they fail to specify the factors that have influenced these waves adequately. While various elements come into play, it is crucial to highlight that lockdown measures were a consequence rather than a trigger factor in managing the pandemic. Additionally, the role of SARS-CoV-2 variants, although significant, must be articulated more coherently in alignment with the paragraph's purpose. Furthermore, the authors should explicitly identify the knowledge gap they aim to address.
Paragraph 3 (Hypothesis and Objectives):
The last paragraph should introduce the paper's hypothesis and objectives, rather than stating something already known: “Each wave exhibits a number of peculiarities”. What was the hypothesis, then? Also, the first sentence of the third paragraph is too long. When a reader reaches the last word of the sentence, he/she has to start over to get the meaning of the hypothesis. We propose the following structure for the objectives:
- First objective: Describe the characteristics of hospitalized COVID-19 patients and assess the virus's impact on mortality and severity.
- Second objective: Identify potential risk factors associated with adverse outcomes in COVID-19 patients.
By employing this clearly defined structure, the authors can effectively convey their objectives to the readers, ensuring better readability and comprehension. Of course, these two statements can be written in accordance with the authors' will.
It seems the authors have difficulties structuring an Introduction section. To improve the overall readability of the Introduction section, the authors must adhere to the suggested revisions, providing a well-structured and coherent framework for their research on the impact of COVID-19.
METHODS
"In the abstract, authors often aim for an economy of words due to word limits, but it is essential to balance brevity with narrative descriptions that engage readers. Readers like to find narrative descriptions. Therefore, instead of the current first sentence, consider starting with: 'This study employs a descriptive, retrospective approach...' to clearly set the tone.
To address the issue of chaotic criteria for including/excluding patients, I recommend creating a flowchart to provide a visual representation of the process. This will enhance the clarity and organization of the information. Additionally, it would be beneficial to differentiate patients hospitalized due to COVID-19 from those with asymptomatic SARS-CoV-2 infections. Was relevant the number of asymptomatic patients with a positive test for SARS-CoV-2 versus a hospitalized patient with severe COVID-19. Exploring the significance of the number of asymptomatic patients with a positive SARS-CoV-2 test compared to hospitalized patients with full-blown COVID-19, respiratory insufficiency, and/or pneumonia would add valuable insights to the study.
In subsection 2.2, the reference to the 'FUENCOVID' local database seems irrelevant to the manuscript's main focus. Unless this database significantly contributes to the study and introduces innovative methods, it might be better suited for a separate publication in a specialized journal.
Regarding subsection 2.3, it's important to clarify that the waves were established by the ISC-III (Instituto de Salud Carlos III) and not the Spanish Ministry of Health. Indeed, the authors cited ISC-III as a reference. But here I find a concern. Waves are defined by the ISC-III based on diagnosed cases, i.e., from the general population in Spain. However, since the definition and limits of waves might vary in a local setting, the authors should explicitly state how they adapted these criteria for their specific population. If a mathematical equation or algorithm was used to define each wave, it should also be clearly explained to ensure transparency and reproducibility. What was the population on which authors based their calculations: the general population in their local area, or hospitalized patients? What was the mathematical equation or algorithm used to define each wave? These criteria should be clearly stated.
In subsection 2.4, variables are not well defined. The authors enumerate several variables, but not all variables are analyzed (vaccination status, disease severity in terms of chest X-ray, categorical consideration of oxygen saturation). My main concern is the definition of outcomes. In a study on COVID-19, I would prefer to read a study on in-hospital mortality rather than mortality 3 months after admission. If the scope of the study is to determine the burden of a single institution, in-hospital mortality should be analyzed. However, if global mortality has to be analyzed, authors have included all causes but described none. The real scope has not been stated previously. Did patients die due to COVID-19? OR did they die due to myocardial infarction within the 3 months after discharge? Any other causes? Results can be questionable and they deserve a discussion as a limitation.
The final subsection should be improved. The description of statistical analyses is insufficient and sparse. It is very generic. The authors mention chi-square and Fisher’s test, but they were not further using them. They mention median and IQR, and the Kruskall-Wallis test, but they reported all values in mean and standard deviation. In the Results section, there are analyses that had not been stated in the Methods section. As a “rule of thumb”, I suggest the authors:
- Descriptive analyses, with Table 1 with all characteristics of the studied cohort. Columns can be waves or can be sex (men/women), and a p-value is optional (but recommendable, as can be considered a univariate analysis).
- Correlation analyses between waves, with ANOVA, Kruskall-Wallis or Poisson regression.
- Correlation as “univariate analyses” with the outcomes (whichever the authors decide to include).
- Multivariate analyses to identify the risk factors, using the outcomes as dependent variables.
The authors seem to give us a glimpse of this structure above in the Results section, but this structure is not fully stated in the Methods section, and both the Methods and the Results sections are misleading and confusing.
P- value has not been stated. I guess it is 0.1 rather than the most common 0.05 (it is legitimate to establish it as 0.1), but it was very confusing and misleading to read the tables without spotting which variable was significant.
Although the authors state they will be using logistic regression, there is no odds ratio in the Results section, but hazard ratio. Where were survival analyses with Cox regression mentioned? I can understand it was an unnoticed mistake, but this concern casts doubt about the reliability of the whole study since it raises the question of whether the authors can perform a multivariate analysis.
RESULTS
My first concerns are introduced in the first figure. The caption is incomplete. As mentioned above, the sum of all minor mistakes is too huge, i.e., the overall “error count” becomes a weighted sum of substantial importance. The authors mention several milestones or landmarks, such as the “begin of vaccination”, or “VOC alpha”, and so on. These milestones are purely speculative. The scope of the manuscript is to describe the burden of COVID-19 in a certain population in a local area, not to analyze the impact of several “milestones” in the pandemic. Should this research was conceived to compare “before VOC” versus “after VOC”, the variable “VOC”, or “vaccination”, or whichever should be stated as a variable in an intervention study. These landmarks are purely informative, but the authors stress their importance in the outcomes in the Introduction section, and once more, in the first figure. What was the purpose of spotting these milestones? There has been more milestone during the pandemic (seasonal holidays, end of the lockdown, opening of schools, and so on).
In subsection 3.2 the authors mention readmissions. Are they relevant to the study? Has been defined the “readmission” variable? How have its criteria been defined? Time to readmission? Cause of readmission?
Tables 1 to 6 are too complex. Their content is sometimes repetitive. I recommend combining them in one or two tables. These tables mostly use absolute values and percentages, and median and standard deviation. It is hard to believe that several variables were normally distributed. Hospital stay? ICU stay? Age? Several local studies (Japan, China, USA, UK, Spain, Sweden) have reported that almost all variables are non-normally distributed.
In subsection 3.2.1 there are mentions to mean age and immigrants’ mean age. The authors state the significance of these results, but the reader does not know the reference p-value nor the mean age of immigrants. I wonder whether the mention to immigrants is useful or significant because the authors did not include this variable in the study except for (not shown) descriptive purposes.
Using ANOVA or Kruskall-Wallis is interesting to estimate how a quantitative dependent variable changes according to the levels of one or more categorical independent variables. ANOVA tests whether there is a difference in the means of the groups at each level of the independent variable. Kruskall-Wallis tests difference in medians. My main concern is that either ANOVA or Kruskall-Wallis test is statistically significant if one or more groups falls outside the range of variation predicted by the null hypothesis (all group means are equal). Which group is different? Along the descriptions of the tables, the authors state that one or more variables are different from the other, but they exclusively are based on the final p-value rather than on pairwise comparisons. ANOVA tells us if there are differences among group means, but not what the differences are. To find out which groups are statistically different from one another, a Tukey’s Honestly Significant Difference (Tukey’s HSD) post-hoc test for pairwise comparisons should be performed.
It is appropriate to include people living with VIH, but CD4 count was never mentioned as a variable.
The description and categorization of oxygen saturation is chaotic and confusing. In the Methods section, there is a mention of the categorization of oxygen saturation, but this variable has not been analyzed or expressed in the tables. What is the purpose of measuring “worst oxygen saturation”? Does it have clinical relevance? ANOVA yields a p-value of <0.001 for oxygen saturation, but it is hard to believe that 92.7, 93.9, 92.8 93.3, and so on are different. Please, see my last commentary on ANOVA and pairwise comparisons.
What are the criteria for defining “low/high” oxygen support? It has not been defined in the Methods section. Also, it seems that defined variables such as nasal cannula, reservoir mask, and so on, are not present in any of the tables.
Table 5 uses mean and standard deviation, but I emphasize that it is difficult to believe that oxygen saturation, CRP, Il-6, and so on were normally distributed. Also, the four laboratory parameters included are confusing. Were they measured on admission? When were they measured? A mean CRP of 11.2 mg/dL is not an elevated measurement (I guess the laboratory limit for normal values is 0-5). What are authors measuring?
Subsections 3.3, 3.4, and 3.5 need to be built from scratch. The analyses need to be performed again. The analyses presented are rather confusing. It is hard to identify the dependent variables and when and what bivariate analyses have been performed.
DISCUSSION
Most of the discussion is speculative. The first part of the Discussion section (paragrapha 1 to 6) tries to describe the waves, but the interpretation of the results is speculative. The authors mention only the causes they consider as relevant, but not proved as such. On the other hand, several other causes proved as relevant are not mentioned (please, see above in a previous comment). The second part of the Discussion section is based on the results of the multivariate regression (I guess it was logistic regression, but hazard ratios are reported). Since analyses should be performed again and interpretation of results should be stated over, I recommend the authors to build from scratch the Discussion section and to avoid speculating on results.
REFERENCES
My only concern in this section is the bad use of references. Some “hard statements” are not referenced, such as “COVID-19 is the disease caused by the novel coronavirus known as SARS-CoV-2”. There is an “abuse” of webpages and UpToDate (I do not think UpToDate is a valid reference, having plenty of medical literature to be used).
Overall, although English is not my native language, the manuscript requires intensive English editing. Some sentences are difficult to understand, and certain connectors or conjunctive words are not used properly or are limited, resulting in poor English. Additionally, there are sentences that are too long, spanning 3 to 4 lines. I highly recommend seeking professional English editing services.
Author Response
Dear reviewer,
We sincerely appreciate your interest in our manuscript. We have carefully addressed your inquiries and incorporated the suggested revisions. In response to your feedback, we agree with most of your recommendations. We believe that thanks to your suggestions, we have improved the quality of the article. We have made significant changes to the introduction, discussion, and references. Hence, we may consider citing you in the acknowledgments for your contribution to the paper. We now remain confident that publication is achievable.
Best regards, the authors
REVIEWER 3
In my opinion, this is an interesting, timely, and necessary manuscript. The study describes and documents the COVID-19 pandemic in the years 2020 and 2021 using a hospitalization database from a local area in Spain. It analyzes severity indicators such as hospital admissions, ICU admissions, and deaths. These indicators are important in pandemic situations and are of great importance for vaccine effectiveness studies. But more importantly, it assesses the impact of the pandemic in terms of hospitalization. Studies like this allow epidemiologists to complement and improve their population databases, although they represent a subset of the information generated at the population level. And they are also useful for monitoring pandemic situations like COVID-19.
However, I have several concerns about it.
GENERAL
Overall, although English is not my native language, the manuscript requires intensive English editing. Some sentences are difficult to understand, and certain connectors or conjunctive words are not used properly or are limited, resulting in poor English. Additionally, there are sentences that are too long, spanning 3 to 4 lines. I highly recommend seeking professional English editing services.
Response: We noticed a discrepancy in the assessment of English language proficiency between the two others reviewers, that deem no corrections necessary. We agree that some long and hard-to-comprehend sentences might be the problem, and we have worked on resolving them. Anyway, we have sent the document to a native English speaker who charged us for their services. We remain confident that publication is achievable.
One notable issue is that the objectives are not clearly stated. In the abstract, it merely states: "Our objective was to study mortality and severity, and associated factors in our hospitalized patients." However, it does not specify the context or focus of this study. For instance, it does not clarify whether the objective is to study mortality and severity in terms of ICU admissions or the need for oxygen support. While reading the manuscript, I formed an overall idea of the objectives, and they seemed different from what was stated in both the abstract and the Introduction section.
Response: we totally agree. We rewrite this section, following your indications
After thoroughly reading the manuscript, and considering it is just my opinion, the first objective appears to be describing the burden of COVID-19 in the author's institution. On the other hand, the second objective seems to be highlighting the association of several variables with mortality and/or severity. Therefore, the proposed study should be categorized as a retrospective, cross-sectional descriptive research. However, I will emphasize later the type of analyses required, as it seems that the authors are not clear on the difference between a cross-sectional study and a longitudinal study. This is evident from their reporting of HR (hazard ratios) instead of OR (odds ratios).
Response: This reviewer proposes a very different approach to our work. Overall, and upon careful consideration of their suggestions, we agree that their perspective is quite accurate and offers a more robust way to extract information from our dataset, which encompasses over 5000 patients. We want to express our gratitude for their valuable insights; we have substantially altered the article's focus based on his (o her) feedback.
INTRODUCTION
The first paragraph should serve as an effective introduction to the main topic, COVID-19. However, the current version is tangled up and chaotic due to excessively long sentences and the use of web page references instead of scientific papers. Therefore, we propose breaking the first paragraph into two or three, focusing on presenting the topic clearly and concisely.
The authors used too long sentences, such as “On March 11, 2020, as the virus progressively spread worldwide, the WHO declared it a pandemic, which on May 05, 2023, was declared over.” Also, this first paragraph cited eight references from web pages. The first eight references should not be web pages, but relevant scientific papers.
Paragraph 2 (Factors Influencing Waves): In the subsequent section (line 50 to line 66), the authors enumerate several disjointed topics and ideas related to the emergence of COVID-19 waves. However, they fail to specify the factors that have influenced these waves adequately. While various elements come into play, it is crucial to highlight that lockdown measures were a consequence rather than a trigger factor in managing the pandemic. Additionally, the role of SARS-CoV-2 variants, although significant, must be articulated more coherently in alignment with the paragraph's purpose. Furthermore, the authors should explicitly identify the knowledge gap they aim to address.
Paragraph 3 (Hypothesis and Objectives):
The last paragraph should introduce the paper's hypothesis and objectives, rather than stating something already known: “Each wave exhibits a number of peculiarities”. What was the hypothesis, then? Also, the first sentence of the third paragraph is too long. When a reader reaches the last word of the sentence, he/she has to start over to get the meaning of the hypothesis. We propose the following structure for the objectives:
- First objective: Describe the characteristics of hospitalized COVID-19 patients and assess the virus's impact on mortality and severity.
- Second objective: Identify potential risk factors associated with adverse outcomes in COVID-19 patients.
By employing this clearly defined structure, the authors can effectively convey their objectives to the readers, ensuring better readability and comprehension. Of course, these two statements can be written in accordance with the authors' will.
It seems the authors have difficulties structuring an Introduction section. To improve the overall readability of the Introduction section, the authors must adhere to the suggested revisions, providing a well-structured and coherent framework for their research on the impact of COVID-19.
Response: we strongly agree with the reviewer that introduction was not very clear. We rewrite this section and add references.
METHODS
"In the abstract, authors often aim for an economy of words due to word limits, but it is essential to balance brevity with narrative descriptions that engage readers. Readers like to find narrative descriptions. Therefore, instead of the current first sentence, consider starting with: 'This study employs a descriptive, retrospective approach...' to clearly set the tone.
Response: done
To address the issue of chaotic criteria for including/excluding patients, I recommend creating a flowchart to provide a visual representation of the process. This will enhance the clarity and organization of the information. Additionally, it would be beneficial to differentiate patients hospitalized due to COVID-19 from those with asymptomatic SARS-CoV-2 infections. Was relevant the number of asymptomatic patients with a positive test for SARS-CoV-2 versus a hospitalized patient with severe COVID-19. Exploring the significance of the number of asymptomatic patients with a positive SARS-CoV-2 test compared to hospitalized patients with full-blown COVID-19, respiratory insufficiency, and/or pneumonia would add valuable insights to the study.
Response: To detect asymptomatic SARS-CoV-2 infections was not our objective and these patients were not collected in our database, so unfortunately we cannot include a flowchart with number of excluded patients because we do not know the number. We try to clarify and included as limitations
In subsection 2.2, the reference to the 'FUENCOVID' local database seems irrelevant to the manuscript's main focus. Unless this database significantly contributes to the study and introduces innovative methods, it might be better suited for a separate publication in a specialized journal.
Response: We use it to collected variables in a “network” system, we think is important because million of data are included, but perhaps is not relevant for study purpose. We erased
Regarding subsection 2.3, it's important to clarify that the waves were established by the ISC-III (Instituto de Salud Carlos III) and not the Spanish Ministry of Health. Indeed, the authors cited ISC-III as a reference. But here I find a concern. Waves are defined by the ISC-III based on diagnosed cases, i.e., from the general population in Spain. However, since the definition and limits of waves might vary in a local setting, the authors should explicitly state how they adapted these criteria for their specific population. If a mathematical equation or algorithm was used to define each wave, it should also be clearly explained to ensure transparency and reproducibility. What was the population on which authors based their calculations: the general population in their local area, or hospitalized patients? What was the mathematical equation or algorithm used to define each wave? These criteria should be clearly stated.
Response: We agree about ISIII, we changed the reference to ministry.
We had the same concern about these point, we apologize because we didn't explain clearly. We try to clarify how we adapted the definition of epidemic waves. Essentially, instead of considering the increase in incidence as the start of a wave, we consider the sustained rise in the number of COVID-admitted patients compared to the previous week in our hospital. No complex algoritm. This scheme resembles that employed by the ISCIII for wave definitions, with the alteration of substituting an increase in incidence with an increase in admissions. In the subanalysis we conducted, no significant differences emerged. However, precisely because we were concerned about this point and fitted better with the hospital's reality, we prefer not to use ISCIII epidemic waves.
In subsection 2.4, variables are not well defined. The authors enumerate several variables, but not all variables are analyzed (vaccination status, disease severity in terms of chest X-ray, categorical consideration of oxygen saturation). My main concern is the definition of outcomes. In a study on COVID-19, I would prefer to read a study on in-hospital mortality rather than mortality 3 months after admission. If the scope of the study is to determine the burden of a single institution, in-hospital mortality should be analyzed. However, if global mortality has to be analyzed, authors have included all causes but described none. The real scope has not been stated previously. Did patients die due to COVID-19? OR did they die due to myocardial infarction within the 3 months after discharge? Any other causes? Results can be questionable and they deserve a discussion as a limitation.
Response: Definitions are in the supplementary data. We added data from X-chest ray, that was mistaken in table 5 and it is very relevant.
About mortality, we believe that this is one of the unresolved questions about COVID-19, and we concur that the approach is debatable. Long covid is considered when symptoms persists for more than 3 months. COVID-19 has the concern, not resolved, of an increased risk of hypercoagulability weeks after discharge. How can we be certain that pulmonary embolisms, strokes, or myocardial infarctions in next weeks after covid admission are unrelated to COVID-19? So we evaluate mortality of “short covid”. We explain in discussion. Unfortunately, we could not documented causes of death in our study.
The final subsection should be improved. The description of statistical analyses is insufficient and sparse. It is very generic. The authors mention chi-square and Fisher’s test, but they were not further using them. They mention median and IQR, and the Kruskall-Wallis test, but they reported all values in mean and standard deviation. In the Results section, there are analyses that had not been stated in the Methods section. As a “rule of thumb”, I suggest the authors:
- Descriptive analyses, with Table 1 with all characteristics of the studied cohort. Columns can be waves or can be sex (men/women), and a p-value is optional (but recommendable, as can be considered a univariate analysis).
- Correlation analyses between waves, with ANOVA, Kruskall-Wallis or Poisson regression.
- Correlation as “univariate analyses” with the outcomes (whichever the authors decide to include).
- Multivariate analyses to identify the risk factors, using the outcomes as dependent variables.
The authors seem to give us a glimpse of this structure above in the Results section, but this structure is not fully stated in the Methods section, and both the Methods and the Results sections are misleading and confusing.
P- value has not been stated. I guess it is 0.1 rather than the most common 0.05 (it is legitimate to establish it as 0.1), but it was very confusing and misleading to read the tables without spotting which variable was significant.
Although the authors state they will be using logistic regression, there is no odds ratio in the Results section, but hazard ratio. Where were survival analyses with Cox regression mentioned? I can understand it was an unnoticed mistake, but this concern casts doubt about the reliability of the whole study since it raises the question of whether the authors can perform a multivariate analysis.
Response: Completely agree, here we have to apologize. We correctly conducted the statistical tests, but we included two major unnoticed mistakes when move to the text. Median was not included in final tables and HR was mentioned, not OR. You can see that OR was correct in the abstract. We solved these and other minor recommendations. We try to explain properly method
RESULTS
My first concerns are introduced in the first figure. The caption is incomplete. As mentioned above, the sum of all minor mistakes is too huge, i.e., the overall “error count” becomes a weighted sum of substantial importance. The authors mention several milestones or landmarks, such as the “begin of vaccination”, or “VOC alpha”, and so on. These milestones are purely speculative. The scope of the manuscript is to describe the burden of COVID-19 in a certain population in a local area, not to analyze the impact of several “milestones” in the pandemic. Should this research was conceived to compare “before VOC” versus “after VOC”, the variable “VOC”, or “vaccination”, or whichever should be stated as a variable in an intervention study. These landmarks are purely informative, but the authors stress their importance in the outcomes in the Introduction section, and once more, in the first figure. What was the purpose of spotting these milestones? There has been more milestone during the pandemic (seasonal holidays, end of the lockdown, opening of schools, and so on).
Response: We appreciate your feedback regarding the graphic. Our intention with the graphic was solely to provide informative visual representation. We would like to point out that a similar graphic highlighting these significant points has been utilized by both references 10 and 16. Our aim was to create a clear and easily comprehensible graph that included some of the most notable pandemic milestones for informational purposes. While including all milestones is impractical, we believe the ones incorporated are indeed relevant, in line with the approach of reference 10. We recognize that the primary objective of our study does not involve analyzing the impact of these "milestones," so the concern might not be exclusively about the graphic but could pertain to the adequacy of references. However, you can read reference 10 that is also very speculative. To address this, we've taken steps to de-emphasize this aspect in both the introduction and discussion, incorporating appropriate references and remarking we can only speculate about milestones.
In subsection 3.2 the authors mention readmissions. Are they relevant to the study? Has been defined the “readmission” variable? How have its criteria been defined? Time to readmission? Cause of readmission?
Response: we refers really to a second episode of covid for the same patient, similar to reference 28. We remark.
Tables 1 to 6 are too complex. Their content is sometimes repetitive. I recommend combining them in one or two tables. These tables mostly use absolute values and percentages, and median and standard deviation. It is hard to believe that several variables were normally distributed. Hospital stay? ICU stay? Age? Several local studies (Japan, China, USA, UK, Spain, Sweden) have reported that almost all variables are non-normally distributed.
Responde: Agree. Changed
In subsection 3.2.1 there are mentions to mean age and immigrants’ mean age. The authors state the significance of these results, but the reader does not know the reference p-value nor the mean age of immigrants. I wonder whether the mention to immigrants is useful or significant because the authors did not include this variable in the study except for (not shown) descriptive purposes.
Response: Agree. Deleted
Using ANOVA or Kruskall-Wallis is interesting to estimate how a quantitative dependent variable changes according to the levels of one or more categorical independent variables. ANOVA tests whether there is a difference in the means of the groups at each level of the independent variable. Kruskall-Wallis tests difference in medians. My main concern is that either ANOVA or Kruskall-Wallis test is statistically significant if one or more groups falls outside the range of variation predicted by the null hypothesis (all group means are equal). Which group is different? Along the descriptions of the tables, the authors state that one or more variables are different from the other, but they exclusively are based on the final p-value rather than on pairwise comparisons. ANOVA tells us if there are differences among group means, but not what the differences are. To find out which groups are statistically different from one another, a Tukey’s Honestly Significant Difference (Tukey’s HSD) post-hoc test for pairwise comparisons should be performed.
Responde: Your insights are highly relevant, and we genuinely appreciate your input. We have incorporated these analyses.
It is appropriate to include people living with VIH, but CD4 count was never mentioned as a variable.
Response: deleted
The description and categorization of oxygen saturation is chaotic and confusing. In the Methods section, there is a mention of the categorization of oxygen saturation, but this variable has not been analyzed or expressed in the tables. What is the purpose of measuring “worst oxygen saturation”? Does it have clinical relevance? ANOVA yields a p-value of <0.001 for oxygen saturation, but it is hard to believe that 92.7, 93.9, 92.8 93.3, and so on are different. Please, see my last commentary on ANOVA and pairwise comparisons.
Response: changed to categorical variable
What are the criteria for defining “low/high” oxygen support? It has not been defined in the Methods section. Also, it seems that defined variables such as nasal cannula, reservoir mask, and so on, are not present in any of the tables.
Response: Definitions are in supplementary material
Table 5 uses mean and standard deviation, but I emphasize that it is difficult to believe that oxygen saturation, CRP, Il-6, and so on were normally distributed. Also, the four laboratory parameters included are confusing. Were they measured on admission? When were they measured? A mean CRP of 11.2 mg/dL is not an elevated measurement (I guess the laboratory limit for normal values is 0-5). What are authors measuring?
Response: Definitions are in supplementary material. Parameters are the worst during admission. Our laboratory report CRP in mg/dL (normal <0.5), We changed to mg/L, more used.
Subsections 3.3, 3.4, and 3.5 need to be built from scratch. The analyses need to be performed again. The analyses presented are rather confusing. It is hard to identify the dependent variables and when and what bivariate analyses have been performed.
Response: scratched and remade
DISCUSSION
Most of the discussion is speculative. The first part of the Discussion section (paragrapha 1 to 6) tries to describe the waves, but the interpretation of the results is speculative. The authors mention only the causes they consider as relevant, but not proved as such. On the other hand, several other causes proved as relevant are not mentioned (please, see above in a previous comment). The second part of the Discussion section is based on the results of the multivariate regression (I guess it was logistic regression, but hazard ratios are reported). Since analyses should be performed again and interpretation of results should be stated over, I recommend the authors to build from scratch the Discussion section and to avoid speculating on results.
Response: scratched and remade. We describe our results, comparing with others (double of references). We remark main findings. After your recommendations, we agree it was improvable.
REFERENCES
My only concern in this section is the bad use of references. Some “hard statements” are not referenced, such as “COVID-19 is the disease caused by the novel coronavirus known as SARS-CoV-2”. There is an “abuse” of webpages and UpToDate (I do not think UpToDate is a valid reference, having plenty of medical literature to be used).
Response: we agree. You can verify that we have substantially increased the number and quality of references.
Comments on the Quality of English Language
Overall, although English is not my native language, the manuscript requires intensive English editing. Some sentences are difficult to understand, and certain connectors or conjunctive words are not used properly or are limited, resulting in poor English. Additionally, there are sentences that are too long, spanning 3 to 4 lines. I highly recommend seeking professional English editing services.
Response: We noticed a discrepancy in the assessment of English language proficiency between the two others reviewers, that deem no corrections necessary. We agree that some long and hard-to-comprehend sentences might be the problem, and we have worked on resolving them. Anyway, we have sent the document to a native English speaker who charged us for their services. We remain confident that publication is achievable.

Round 2
Reviewer 2 Report
The paper works well. Please review the style of the document, referring to Author guidelines, review the references (there are several mistakes), improve the section about the contributions, funding, etc.
The paper is well-written